# WalK(S221P) Mutation Promotes the Production of *Staphylococcus aureus* Capsules Through an MgrA-Dependent Pathway

**DOI:** 10.3390/microorganisms13030502

**Published:** 2025-02-25

**Authors:** Zuwen Guo, Huagang Peng, Weilong Shang, Yi Yang, Zhen Hu, Yifan Rao, Xiaonan Huang, Jianxiong Dou, Zihui Xu, Xiancai Rao

**Affiliations:** 1Key Laboratory of Microbial Engineering under the Educational Committee in Chongqing, Department of Microbiology, College of Basic Medical Sciences, Army Medical University (Third Military Medical University), Chongqing 400038, China; guozuwen6@sina.com (Z.G.); huagangpeng@tmmu.edu.cn (H.P.); shangwl@tmmu.edu.cn (W.S.); yangyi980783@tmmu.edu.cn (Y.Y.); huzhenzhen1314@tmmu.edu.cn (Z.H.); huangxn1999@126.com (X.H.); 17815352663@163.com (J.D.); 2Department of Emergency Medicine, Xinqiao Hospital, Army Medical University (Third Military Medical University), Chongqing 400037, China; raoyifan1995@tmmu.edu.cn; 3Department of Integrative Medicine, Xinqiao Hospital, Army Medical University (Third Military Medical University), Chongqing 400037, China

**Keywords:** *Staphylococcus aureus*, virulence, capsular polysaccharide, WalKR, two-component system, MgrA

## Abstract

*Staphylococcus aureus* is a vital pathogen causing clinical infections. Capsules are important virulence factors for *S*. *aureus*. This study investigates the regulatory mechanisms underlying capsule production in *S. aureus*. Bacterial strains XN108 and Newman were used, and combined approaches like RNA sequencing (RNA-seq), RT-qPCR, transmission electron microscopy (TEM), gene reporter, and electrophoretic mobility shift assay (EMSA) were performed to test the role and mechanism of WalK(S221P) mutation in *S. aureus* capsule production. RNA-seq showed an increased expression of *cap* genes in the WalK(S221P)-carried *S. aureus* XN108 relative to the mutation-cured XN108-R. TEM and capsular polysaccharide determination demonstrated that XN108 produced more capsules than XN108-R did. Similar results were presented in the WalK(S221P)-contained K-Newman versus the wild-type Newman. RT-qPCR screening showed an increasing expression of the *mgrA* gene in XN108 versus XN108-R. Gene reporter and EMSA analysis revealed that WalK(S221P) mutation promoted *S. aureus* capsule production through MgrA. Deletion of *mgrA* decreased the WalK(S221P)-mediated capsule yield. Moreover, WalK(S221P) mutation remarkably increased the tolerance of *S. aureus* to whole blood killing and microphage phagocytosis. Overall, these data provide mechanistic insights into the effect of WalK(S221P) on the capsule production of *S. aureus*, which may set down foundations for future *S. aureus* virulence investigations.

## 1. Introduction

*Staphylococcus aureus* (*S. aureus*) is an important pathogen that can cause a variety of diseases, including skin/soft tissue disorders, pneumonia, myocarditis, and sepsis [1]. *S. aureus* often results in the high mortality of patients suffering with influenza or SARS-CoV-2 infections [2,3]. According to a report from the World Health Organization, there are approximately 20,000 *S. aureus* bacteremia-related deaths in the United States each year [4]. In China, *S. aureus* is ranked as the most clinically important Gram-positive bacteria according to the China Antimicrobial Surveillance Network (http://www.chinets.com, accessed on 23 December 2024). Virulence factors are morbigenous weapons used by pathogenic bacteria to cause diseases [5]. The notorious *S. aureus* can secrete plenty of virulence factors, such as enterotoxins, hemolysins, exfoliative toxins, leucocidins, proteases, and toxic shock syndrome toxin 1 (TSST-1) [6,7,8]. Capsules are constituted with diverse capsular polysaccharides (CPs) and also important virulence factors for *S. aureus* [9]. *S. aureus* capsules can protect bacterial cells from phagocytosis and facilitate bacteria to escape host immune inactivation [10]. To date, 11 serotypes of *S. aureus* CPs have been categorized [11]. However, more than 90% of clinically isolated *S. aureus* strains produce type 5 (CP5) or type 8 CPs (CP8) [12,13].

The production of *S. aureus* capsules is controlled by a *cap* operon, which has 16 genes (*capA*–*capP*) encoding various enzymes involved in CP biosynthesis and transportation [14]. The expression of the *cap* gene cluster is strictly controlled during *S. aureus* growth, showing a repression before the exponential phase, a priming expression in the late exponential stage, and a maximum production at the stationary phase [15]. This timely expression profile of *S. aureus cap* genes may be ascribed to a regulatory network that governs the *cap* promoter activity. Recently, Keinhörster et al. have revisited the regulation of the *cap* gene cluster in *S. aureus* and found that the promoter element of *cap* operon (P*cap*) contains a major sigma B (SigB)-dependent promoter (P*sigB*) and a weak SigA-dependent promoter (P*sigA*) [16]. An array of positive regulators such as MgrA, SpoVG, and RbsR and negative modulators like Rot, CodY, and SaeSR control the expression of the *cap* gene cluster by directly affecting P*sigB* or P*sigA* activities [16,17,18,19]. MgrA is a small transcriptional modulator that affects the expression of a large number of genes [20,21]. Lei and Lee reported that MgrA directly binds to the P*cap* and affects the SigA-dependent promoter in *S. aureus* [19]. Other regulators such as SigB, SbsDC, and the two-component systems (TCS) AgrCA and ArlSR can influence CP biosynthesis through MgrA [20,22,23,24]. WalKR is the only essential TCS for *S. aureus* survival by regulating bacterial cell wall metabolism [25]. The expression of approximately 200 genes in *S. aureus* is directly and indirectly regulated by WalKR [26]. However, whether WalKR affects CP production in *S. aureus* has not been determined.

In the previous studies, we characterized a vancomycin-intermediate *S. aureus* (VISA) strain XN108 and found that a TCS element mutation, WalK(S221P), is associated with the vancomycin resistance and virulence of XN108 [26,27,28,29,30]. The present study aims to investigate the role and mechanism of WalK(S221P) mutations in *S. aureus* capsule production. RNA sequencing (RNA-seq) and RT-qPCR analysis showed that the WalK(S221P) mutation resulted in the upregulation of *S. aureus cap* genes. Transmission electron microscopy (TEM) and CP quantitative determination showed that the WalK(S221P) mutation-carried *S. aureus* strains XN108 and K-Newman produced more CPs than their counterparts. Further, the LacZ gene reporter, the electrophoretic mobility shift assay (EMSA), and CP observation and quantitation revealed that WalK(S221P) mutation promoted capsule production through an MgrA-dependent pathway. The inactivation of *mgrA* impaired the WalK(S221P)-mediated CP yield and, when complemented with *mgrA*, restored the CP biosynthesis. Moreover, WalK(S221P) mutation increased the survival of *S. aureus* in the whole-blood inactivation and after phagocytosis by microphages. Overall, our data presented a WalKR-MgrA axis that controls the capsule production in *S. aureus*, and the WalK(S221P) mutation may enhance the CP yield through the upregulation of *mgrA* expression.

## 2. Materials and Methods

### 2.1. Bacterial Strains and Plasmids

The bacterial strains and plasmids used in this study are listed in Table 1. The methicillin-resistant *S. aureus* (MRSA) and VISA strain XN108 were isolated from a burn patient suffering from an acute *S. aureus* infection [27]. The methicillin-susceptible *S. aureus* (MSSA) strain Newman (NCTC 8178) was provided by Prof. Lu Yu (Jilin University). XN108-R and K-Newman were constructed as described previously [26]. A pLI50 plasmid was used for gene complementation, plasmid pOS1 was used to construct reporter, pET-28a was used to prepare recombinant proteins, and *Escherichia coli*-*S. aureus* shuttle vector pBT2 was used for gene deletion. *S. aureus* and its derivative strains were cultured in a brain heart infusion (BHI, Oxoid, Hampshire, UK), and chloramphenicol (20 μg/mL) was added when necessary. *E. coli* strains were grown in a Luria–Bertani (LB) medium with ampicillin (100 μg/mL) or kanamycin (50 μg/mL) as needed.

### 2.2. RNA-Seq

*S. aureus* cells XN108 or XN108-R at the logarithmic phase were harvested, and the total RNA was prepared with an RNAprep Pure Cell/Bacteria Kit (TianGen, Beijing, China). Three biological replicate samples were used for RNA-seq, and the cDNA libraries were generated from each RNA sample with a TruSeq Illumina Kit (San Diego, CA, USA) as described [29]. RNA-seq was performed in Novogene Co., Ltd. (Beijing, China) with the Illumina HiSeqTM2500 sequencing system. The analysis of the results was performed according to the protocol recommended by the manufacturer (Illumina Inc., San Diego, CA, USA). The software DESeq version 1.10.1 was used to analyze differentially expressed genes (DEGs), as previously described [26]. The RNA-seq raw data were submitted and deposited in the GEO DataSets with an ID code of GSE127706.

### 2.3. RT-qPCR

*S. aureus* strains XN108 and XN108-R were cultured in a BHI at 37 °C for 12 h. Total RNA from each strain were extracted with the RNAprep Pure Cell/Bacteria Kit (TianGen, China). The relevant cDNA samples were obtained using the RevertAid First Strand cDNA Synthesis Kit (Thermo Scientific, Waltham, MA, USA). Afterward, the quantitative polymerase chain reaction (qPCR) was conducted to detect the relative expression level of each target gene to the reference *gyrB*, as previously described [26]. The primers used in this study were provided in Appendix A.

### 2.4. CP Extraction and Quantitation

The CP extraction was performed as previously described with some modifications [24]. Briefly, the *S. aureus* strains were cultured in a BHI at 37 °C for 2 d. Bacterial cells were harvested by centrifugation at 8000× *g* for 10 min, resuspended in phosphate-buffered saline (PBS, pH 7.2), autoclaved at 121 °C for 50 min, and then centrifuged at 8000× *g* for 10 min. The supernatant was filtered through a 0.22 μm membrane (Millipore, Burlington, MA, USA) and kept at 4 °C overnight. Precooled ethanol (4 °C) was added to a final concentration of 80% (*v*/*v*) and incubated overnight at 4 °C to precipitate the CPs. After washing twice with precooled 80% (*v*/*v*) and once with 96% (*v*/*v*) ethanol, the CPs were collected by centrifugation at 15,000× *g* for 10 min, dissolved in PBS, dialyzed in ddH_2_O for 2 d, and finally lyophilized. Then, the lyophilized sample was dissolved in buffer A (2 mM MgSO_4_, 50 mM Tris-HCl, pH 7.5) to a concentration of 10 mg/mL and treated with DNase I (Solarbio, Beijing, China) and RNase A (Sigma, St. Louis, MO, USA) at 1 mg/mL at 37 °C for 6 h. Subsequently, the mixture was treated with 2 mg/mL Protein K (Merck Millipore, Darmstadt, Germany) at 65 °C for 6 h. After centrifugation at 15,000× *g* for 15 min, the supernatant was ultrafiltered using a 30 kDa MWCO centrifugal tube (Merck Millipore) and lyophilized. The lyophilized material was dissolved in PBS to a concentration of 20 mg/mL, and 0.05 M NaIO_4_ was added to remove teichoic acids. The resulting mixture was kept in dark conditions until the value of optical density at 233 nm (OD_233_) was stable. The reaction was stopped by adding 10% (*v*/*v*) 16 M ethylene glycol. Ultrafiltration with a 100 kDa MWCO centrifugal tube was performed to collect the filtered solution, which was further ultrafiltered with a 50 kDa MWCO centrifuge tube to obtain the CP solution. After dialysis in ddH_2_O for 2 d, the CP sample was lyophilized and kept at −80 °C.

### 2.5. TEM Observation

To analyze the morphology of the capsule structure, *S. aureus* strains of interest were cultured on Columbia agar plates supplemented with 2% (*m*/*v*) NaCl (Thermo Scientific, Waltham, MA, USA) at 37 °C for 24 h. Bacterial cells were collected and washed with PBS. Then, the cell sample was prepared with a method of lysine acetate-based formaldehyde/glutaraldehyde ruthenium red osmium fixation procedure, as previously described [32]. In brief, *S. aureus* cells were fixed with 2% (*v*/*v*) formaldehyde and 2.5% (*v*/*v*) glutaraldehyde in a cacodylate buffer supplemented with 0.075% (*m*/*v*) ruthenium red and 0.075 M lysine acetate on ice for 20 min. After washing with the cacodylate buffer containing ruthenium red, bacteria were secondly fixed with formaldehyde and glutaraldehyde for 3 h, followed by fixation with 1% (*m*/*v*) osmium in the cacodylate buffer with ruthenium red for 1 h at room temperature. Washing was performed five times with the ruthenium red-carrying cacodylate buffer, and bacterial samples were then dehydrated with gradient concentrations of ethanol (10, 30, 50, 70, 90, and 100%) on ice for 30 min per step. After infiltration with the acrylic resin (London Resin Company, London, UK), the ultrathin sections were prepared. Finally, bacterial sample sections were observed under a TECNAI 10 transmission electron microscope (Philips, The Netherlands) and photographed.

### 2.6. β-Galactosidase Assay

The DNA fragments carrying the *mgrA*-promoter (P*mgrA*) and *cap*-promoter (P*cap*) were obtained by PCR using primer pairs pOS-*mgrA*-F/R and pOS-*cap*-F/R (Appendix A), respectively. Subsequently, PCR products were digested with *Bam*H I and *Eco*R I and cloned into pOS1 plasmid to generate pOS-*mgrA* and pOS-*cap*. The resulting plasmids were transformed into *S. aureus* RN4220 for restriction modification, followed by transformation into XN108, XN108-R, and XN108Δ*mgrA*.

The β-galactosidase assay was conduct as described [20]. In brief, *S. aureus* strains transformed with pOS-*mgrA* or pOS-*cap* were cultured in a BHI medium supplemented with chloramphenicol (10 μg/mL) at 37 °C overnight. On the next day, the culture was inoculated in a fresh BHI broth (1:200 dilution) and cultivated at 37 °C for 8 h. Bacterial cells were harvested by centrifugation at 10,000× *g* for 1 min and resuspended in an AB buffer (100 mM NaCl, 100 mM KH_2_PO_4_, pH 7.0) with consistent OD_600_ values. For each reporter strain, 10 μL of the bacterial solution was added into 85 μL of the AB buffer, and lysostaphin (Sigma, St. Louis, MO, USA) at a final concentration of 5 μg/mL was added to the lyse bacterial cells by incubating at 37 °C for 2 h. Then, 10 μL of the mixture was transferred into a well of 96-well plates with 15 μL of 1% (*v*/*v*) Triton X-100. Subsequently, 30 μL of 4-methylumbelliferyl-β-d-galactoside (Sigma, St. Louis, MO, USA) was added at a final concentration of 1 mg/mL, and the plate was incubated at 37 °C for 1 h. Finally, the reaction was stopped by adding 75 μL of a stop buffer (15 mM EDTA, 300 mM glycine, pH 11.2). The fluorescence intensity was measured using an excitation wavelength of 360 nm and an emission wavelength of 460 nm.

### 2.7. Protein Expression and Purification

The recombinant proteins WalK and WalR were prepared as described [16]. To obtain recombinant MgrA, its gene was amplified by PCR with primers PET28a-*mgrA*-F/R (Appendix A) from the genomic DNA of *S. aureus* XN108. A 6 × His tag was introduced by primer PET28a-*mgrA*-R and fused with the C-terminal of MgrA protein. The PCR product was digested with *Bam*H I and *SaI* I and cloned into a pET28a vector to generate pET-mgrA. Subsequently, the vector was transformed into *E. coli* strain BL21(DE3), and 0.1 μM isopropyl-β-d-thiogalactoside (IPTG) was used to induce the expression of MgrA. The MgrA proteins were purified as previously described [16]. The purified protein was quantified using the Bradford protein assay kit (Bio-Rad, Hercules, CA, USA) and stored at −80 °C until use.

### 2.8. EMSA

The biotin-labelled DNA probes of the *cap* (P*cap*), *mgrA* (P*mgrA*), and *agr* (*agr*) gene promoters were obtained from *S. aureus* XN108 genomic DNA by PCR with the primers *cap*-F-biotin/*cap*-R, *mgrA*-F-biotin/*mgrA*-R, and *agr*-F-biotin/*agr*-R, respectively (Appendix A). An unrelated DNA fragment *hu* was also amplified by PCR using the primer pair *hu*-F/*hu*-R to serve as a nonspecific control. The unlabeled DNA fragment amplified with the primer pair *mgrA*-F/*mgrA*-R or *cap*-F/*cap*-R was used in the binding competition. The EMSA was performed as previously described [16]. Briefly, 0.25 nM of P*cap* or P*mgrA* probes was added to various amount of WalR (0 to 8 μM) activated with 1 μM of WalK and 5 mM of ATP in 20 μL of a reaction buffer containing 1 mM EDTA, 100 mM NaCl, and 50 mM Tris−HCl at a pH of 8.0 [23]. For MgrA binding to P*cap*, 0.5 nM of P*cap* DNA was added to diverse concentrations of MgrA (0 to 4 μM) in 20 μL of the reaction buffer containing 20 mM HEPES (pH 8.0), 1 mM DTT, 100 mM NaCl, 1% (*v*/*v*) glycerol, and sheared salmon sperm DNA (3 ng/μL), as described [20]. After incubation at 25 °C for 30 min, the reaction mixture was separated on 6% (*m*/*v*) natural acrylamide gels and electrophoresed with a 0.5 × TBE buffer at 100 V for 120 min. Subsequently, the samples were electrotransferred onto a positively charged nylon membrane (Beyotime, Shanghai, China), cross-linked with UV, and developed with a Chemiluminescent Nucleic Acid Detection Kit (Thermo scientific). The image was obtained using a Chemiluminescence imager (VILBER, Collégien, France).

### 2.9. Genetic Construction

The gene deletion and complementary *S. aureus* strains were constructed as previously described [26]. Taking the generation of XN108Δ*mgrA* as an example, about 1000 bp DNA fragments flanking left and right DNA sequences of the *mgrA* gene were amplified from XN108 genomic DNA by PCR with the primer pairs pBT2-*mgrA*-left-F/*mgrA*-left-R, and *mgrA*-right-F/pBT2-*mgrA*-right-R (Appendix A), respectively. The left and right DNA fragments were fused by over-lap PCR and cloned into pBT2 to obtain pBT2Δ*mgrA.* After modification in *S. aureus* RN4220, the pBT2Δ*mgrA* was electroporated into the *S. aureus* strain XN108. The seamless *mgrA* deletion mutant XN108Δ*mgrA* was selected via homologous recombination and verified by PCR and DNA sequencing.

For *mgrA* complementation, the *mgrA* gene fragment with its promoter region was amplified by PCR using primers pLI/*mgrA*-F/R (Appendix A) and cloned into the shuttle plasmid pLI50 to achieve pLI-*mgrA*. After modification, the plasmid was electroporated into XN108Δ*mgrA* to obtain a XN108Δ*mgrA*/pLI-*mgrA* complemented strain.

### 2.10. Bacterial Growth Curve

The growth curves of *S. aureus* XN108 and its derivatives were detected as described [33]. Briefly, bacteria were cultured overnight at 37 °C in a BHI medium with shaking, and the 200 μL overnight culture was inoculated into 20 mL of the fresh BHI broth (1:100 dilution) in a sterile flask (50 mL). The OD_600_ values were determined every hour for 24 h, and the growth curve was drawn by using OD_600_ values throughout the culture time.

### 2.11. Whole Blood Killing

Overnight cultures of *S. aureus* strains were diluted 1:100 into the fresh BHI and grown at 37 °C for 6 h. Bacterial cells were collected by centrifugation, washed twice, and diluted in PBS to adjust to an OD600 of 0.5 (about 10^8^ CFU/mL^−1^). Approximately 10^6^ colony forming unit (CFU) cells in 100 µL PBS were mixed with 900 µL of fresh Sprague–Dawley mouse blood (SCBS, Shanghai, China). The tubes were incubated at 37 °C with slow rotation. Approximately 200 µL mixture was taken after 1 or 2 h of incubation, and 0.5% (*m*/*v*) saponin was used to lyse blood cells. The samples were then serially diluted for bacterial counting with BHI agar plates. The survival rate (%) was calculated as (CFU_time point_/CFU_initial input_) × 100.

### 2.12. Bacterial Survival After Macrophage Phagocytosis

RAW264.7 macrophages were cultured in high-glucose Dulbecco’s modified Eagle’s medium (DMEM, Thermo Scientific, Waltham, MA, USA) supplemented with 10% (*v*/*v*) fetal bovine serum at 37 °C with 5% CO_2_. *S. aureus* cells at the exponential growth phase were collected and resuspended in DMEM. RAW264.7 cells were infected with the *S. aureus* strain at a multiplicity of infection (MOI) of 10 in 24-well plates. At 1 h post infection (hpi), the supernatant was removed, and the macrophages were washed with sterile PBS. About 1 mL DMEM containing lysostaphin (1 mg/mL) and gentamicin (50 µg/mL) was added to eliminate extracellular bacteria by incubating at 37 °C for 1 h. Macrophages were then cultured for 4 or 24 h at 37 °C with 5% CO_2_. After lysis with PBS containing 1% Triton X-100, the bacterial CFUs after phagocytosis were counted with BHI agar plates. The experiment was repeated three times.

### 2.13. Statistical Analysis

The experimental data were analyzed using GraphPad Prism 9.3.1. Data from two independent groups were treated with unpaired two-tailed student’s *t*-tests, and a *p* value less than 0.05 was considered statistically significant.

## 3. Results

### 3.1. WalK(S221P) Mutation Enhances Capsule Production of S. aureus

RNA-seq has been performed to explore the role of WalK(S221P) mutation, and the results showed that the expression levels of all 16 genes involved in the CP biosynthesis of *S. aureus* were remarkably upregulated in XN108 compared with those in the WalK(S221P)-reverted strain XN108-R (Figure 1A). This phenomenon was further confirmed by the RT-qPCR detection of *capA*, *capG*, and *capL* gene expression levels in XN108 versus XN108-R, and a similar expression profile was revealed (Figure 1B). We next detected whether the upregulated *cap* genes after WalK(S221P) mutation can increase capsular production in *S. aureus*. The CPs of XN108 and XN108-R were extracted and quantified with a phenol–sulfuric acid method as described [34,35]. As shown in Figure 1C, *S. aureus* XN108 produced more CPs [(116.1 ± 3.8) µg/g cell weight] than XN108-R [(103.2 ± 1.7) µg/g] did (*p* < 0.01). Moreover, TEM revealed a significantly thicker capsule of XN108 [(39.3 ± 16.7) nm] in comparison to XN108-R [(19.5 ± 8.2) nm, *p* < 0.001, Figure 1D,E]. These results demonstrated that the WalK(S221P) mutation has a positive regulatory effect on the capsule synthesis of the *S. aureus* strain XN108.

Most clinical *S. aureus* isolates carry a CP5 or CP8 capsule [36]. The *S. aureus* strain Newman (CP5) was used to detect whether the observed role of WalK(S221P) mutation in the capsular production of XN108 (CP8) occurs in CP5 isolates. An allelic replacement strain was generated by substituting *walK* in the Newman with that from XN108 (*walk(S221P)*), namely K-Newman [26]. RT-qPCR detection exhibited that the expressions of the *capA*, *capG*, and *capL* genes were considerably increased in K-Newman compared with those in the wild-type Newman (Appendix A). Accompanying the increased *cap* gene expression, K-Newman presented a thickened capsule relative to the wild-type Newman (Appendix A). Overall, these results indicated that WalK(S221P) mutation can promote the production of *S. aureus* capsules.

### 3.2. WalKR Controls Cap Expression Indirectly

To determine whether WalKR controls the expression of *cap* genes, a reporter carrying *lacZ* gene controlled by the *cap* promoter (pOS-*cap*) was constructed, and a β-galactosidase assay was performed after the transformation of pOS-*cap* into *S. aureus* strains XN108 and XN108-R. The results showed that β-galactosidase activity in XN108 was significantly increased compared with that in the XN108-R (*p* < 0.05, Figure 2A), suggesting a positive effect of WalK(S221P) mutation on the promoter activity of *cap* gene cluster. The response regulator WalR executes its role by binding a consensus motif (5′-TGTWAH-N5-TGTWAH-3′) located upstream of the target genes in *S. aureus* [25]. However, no predicted consensus motif of WalR was identified in the promoter regions of *cap* operon in *S. aureus* XN108. To further verify the direct regulatory function of WalR on *cap* operon, the His-tagged recombinant proteins WalK and WalR were prepared (Figure 2B). The EMSA revealed that the WalK-phosphorylated WalR did not bind the biotin-labelled *cap* gene promoter probe but bound to the biotin-labelled *agr* gene promoter probe, which served as a positive control (Figure 2C). Collectively, these data showed that WalKR may indirectly control the expression of *cap* genes.

### 3.3. WalK(S221P) Upregulates Cap Gene Expression Through MgrA

To detect the molecule that mediates the effect of WalK(S221P) mutation on *cap* gene expression, RT-qPCR was performed to determine the altered regulatory genes in *S. aureus* with WalK(S221P) mutation. The results showed that the expression of *mgrA* was mostly increased in XN108 compared with that in XN108-R, followed by *spoVG*, whereas other detected regulators did not change their expression levels in XN108 versus XN108-R (Figure 3A). To further detect whether WalK(S221P) mutation affects *mgrA* expression, a reporter plasmid (pOS-*mgrA*) with the *mgrA* gene promoter-*lacZ* fusion was constructed. After transformation into XN108 and XN108-R, LacZ activity analysis revealed that WalK(S221P)-carrying XN108 exhibited remarkably stronger *mgrA* promoter activity than XN108-R did (Figure 3B). The EMSA showed that WalK-phosphorylated WalR could specifically bound to the *mgrA* promoter DNA probe (Biotin P*mgrA*) in a dose-dependent manner (Figure 3C). The binding was outcompeted by the 300-fold addition of unlabelled P*mgrA* probe DNA; however, the 300-fold addition of an unrelated *hu* DNA fragment did not inhibit the binding. These data demonstrated that WalKR can directly control *mgrA* expression, and WalK(S221P) mutation resulted in the upregulation of *mgrA* in XN108.

MgrA is reportedly used to regulate *cap* gene expression [19]. Thus, a mutant with *mgrA* deletion was constructed (Appendix A). *S. aureus* strains XN108 and XN108Δ*mgrA* were, respectively, transformed with pOS-*cap* plasmid to test the effect of MgrA on *cap* expression. As shown in Figure 3D, the LacZ assay revealed that the activity of the *cap* promoter in XN108 was stronger than that in the XN108Δ*mgrA*, indicating a positive role of MgrA on *cap* gene expression. Next, the recombinant MgrA protein was prepared (Figure 3E), and the EMSA showed that MgrA could bind to the *cap* gene promoter DNA in a dose-dependent manner (Figure 3F). The binding ability of MgrA was outcompeted by the 300-fold addition of unlabeled DNA fragments of P*mgrA*, but not the irrelevant DNA fragment *hu* (Figure 3F). Collectively, these results showed that MgrA directly controls *cap* gene expression, and WalK(S221P) mutation promoted capsule production in *S. aureus* through MgrA.

### 3.4. Deletion of mgrA Reduces Capsule Production

To determine whether the MgrA-mediated effect of WalK(S221P) mutation on *cap* gene expression influences *S. aureus* capsule production, a complementary strain was constructed by the transformation of a pLI*mgrA* expression plasmid into XN108Δ*mgrA.* The deletion and complementation of *mgrA* did not affect the growth of the bacteria XN108Δ*mgrA* and XN108Δ*mgrA*/pLI*mgrA* (Figure 4A). TEM observed that the capsular thickness of XN108Δ*mgrA* [(10.0 ± 7.7) nm] substantially decreased compared with that of the wild-type strain [(39.3 ± 16.7) nm], and complementation with *mgrA* partially increased the capsular thickness of XN108Δ*mgrA*/pLI*mgrA* [(18.3 ± 7.8) nm, Figure 4B,C]. The quantification analysis revealed significantly reduced CPs in XN108Δ*mgrA* [(24.6 ± 1.5) µg/g] and elevated capsular production in XN108Δ*mgrA*/pLI*mgrA* [(107.9 ± 8.6) µg/g], comparable to that of the wild-type XN108 [(116.1 ± 3.8) µg/g, Figure 4D]. These results showed that *mgrA* plays an important role in the promotion of capsule biosynthesis in *S. aureus* with WalK(S221P) mutation.

### 3.5. WalK(S221P)- and MgrA-Mediated Capsule Production Promotes Bacterial Intracellular Survival

*S. aureus* capsules can protect bacteria from phagocytosis, thereby enhancing bacterial intracellular survival [37,38,39]. Neutrophiles that circulate in the blood can inactivate bacteria by oxidative killing [40]. Therefore, whole-blood killing was performed to determine if the enhanced capsule contributes to *S. aureus* survival. The results exhibited that the survival rate of XN108-R and XN108Δ*mgrA* after 1 or 2 h incubation with Sprague–Dawley mouse blood considerably decreased compared with that of XN108 (Figure 5A). Macrophages are the first line of host defence against invading bacteria [41]. To detect whether WalK(S221P)-promoted capsule production contributes to *S. aureus* survival within macrophages, we assessed the survival ability of XN108 or its derivatives within RAW264.7 macrophages. As shown in Figure 5B, XN108-R and XN108Δ*mgrA* remarkably reduced their survival numbers within RAW264.7 macrophages compared with the wild-type XN108 after 24 h of culture. However, comparable survivals of XN108-R, XN108Δ*mgrA*, and XN108 were revealed after 4 h of culture within RAW264.7 macrophages. Overall, these data demonstrated that WalK(S221P) mutation and MgrA promoted the tolerance of *S. aureus* cells to whole blood- and macrophage-mediated killing by upregulating the production of bacterial capsules.

## 4. Discussion

*S. aureus* is a versatile pathogen equipped with an array of virulence factors, which facilitate bacterial colonization, infection, and immune evasion [7,8]. As one of the important immune escape molecules, the capsule plays a vital role in *S. aureus* infections [12]. The regulation of capsule production in *S. aureus* is complex, and many regulators can affect the expression of the *cap* gene cluster directly or indirectly [17,18,19,22,24,42]. WalKR is an essential TCS of *S. aureus* and is responsible for the regulation of cell wall metabolism, protein biosynthesis, nucleotide metabolism, and DNA replicative initiation [25,43]. The previous studies have demonstrated that WalK(S221P) mutation contributed to VISA phenotypes, such as increased vancomycin resistance and attenuated bacterial virulence [22,26]. However, the effects of WalKR mutations on *S. aureus* capsule production are unclear. In the present study, we demonstrated that the WalK(S221P) mutation promoted the capsule production of *S. aureus* by upregulating *mgrA* expression. Consistently, the isogenic deletion of *mgrA* decreased CP biosynthesis, indicating crucial roles of WalK(S221P) and MgrA in modulating the immune escape system in *S. aureus*.

Capsule production is determined by the expression of the cap gene cluster, which encodes the required enzymes for CP biosynthesis [14]. RNA-seq analysis and RT-qPCR detection exhibited that WalK(S221P) mutation resulted in upregulated *cap* gene expression in the CP8 strain XN108 compared with the WalK(S221P)-reverted counterpart XN108-R. This phenomenon was verified in the CP5 strain K-Newman versus wild-type Newman. WalKR comprises a sensor histidine kinase WalK and a DNA-binding response regulator WalR [44]. Upon activation by certain signals, the H385 of WalK is auto-phosphorylated, and the phosphoryl group is then transferred to the aspartate residue 53 (D53) of WalR, which functions as a transcriptional regulator [25]. Therefore, we generated a LacZ reporter, and the β-galactosidase assay revealed a positive effect of WalK(S221P) mutation on the promoter activity of the *cap* gene cluster, whereas the EMSA showed no direct binding of WalK-activated WalR to the *cap* promoter DNA probe, indicating an indirect role of WalKR in *cap* gene expression.

RT-qPCR was performed to screen the potential molecules in mediating the WalKR regulation of *cap* gene cluster. The results demonstrated the expression of *mgrA* was mostly upregulated in the WalK(S221P) mutation-carrying strain. MgrA is a member of the MarR (multiple antibiotic resistance regulator) protein family, which can regulate the expression of virulence factors, including alpha-toxin, coagulase, protein A, nuclease, extracellular serine proteases, and capsule [21]. Luong et al. reported that the ArlRS TCS modulated *S. aureus* capsule production through MgrA [20]. Crosby et al. demonstrated that phosphorylated ArlR bound to the promoter regions to activate *mgrA* expression [45]. We showed that WalK-activated WalR could bind to the *mgrA* promoter DNA probe and MgrA could directly bind to the P*cap* probe. Thus, WalKR and MgrA orchestrated a WalKR-MgrA axis to control the capsule production of *S. aureus*. Structurally, WalK(S221P) mutation occurs in the HAMP (histidine kinases, Adenylylcyclases, Methyl binding proteins, and Phosphatases) domain of WalK, which connects the extracellular sensory and intracellular signalling domains (Figure 6). In a previous study, the in vitro phosphorylation assay demonstrated that WalK(S221P) mutation presented decreased autophosphorylation, which affected the activation of WalR [28]. WalKR may negatively control *mgrA* expression, and the WalK(S221P) mutation reduced the effect of WalKR on the *mgrA* promoter, thereby upregulating *mgrA* to enhance CP biosynthesis in *S. aureus*. However, the fine regulatory mechanism of the WalKR-MgrA axis requires further investigation, which will guide our next work.

As an immune escape molecule, WalK(S221P)-mediated capsule production is supposed to enhance bacterial survival [7,46]. In this study, the whole blood killing and macrophage phagocytosis confirmed this conclusion. The survival rate of XN108-R after incubation with mouse whole blood for 1 or 2 h substantially reduced compared with that of XN108, and a similar result was revealed in the case of XN108Δ*mgrA*. Consistently, the survival numbers of XN108-R and XN108Δ*mgrA* remarkably decreased after 24 h of phagocytosis by RAW264.7 macrophages when compared with that of XN108. These results further confirmed that MgrA mediated the promotion of WalK(S221P) mutation to *S. aureus* capsule production. Therefore, enhancing WalKR activity and MgrA inhibition could be promising strategies to inhibit CP production and facilitate the clinical control of *S. aureus* infections.

## 5. Conclusions

We demonstrated the role of WalK(S221P) mutation in promoting the capsule production of *S. aureus*. Moreover, WalK(S221P) increased CP biosynthesis through MgrA, a member of the MarR protein family, which is a first-line regulator governing *cap* gene expression (Figure 6). In addition, WalK(S221P)-mediated capsule yield contributed to *S. aureus* survival. Our study provided a WalKR-MgrA axis in controlling CP biosynthesis in *S. aureus*, which may establish a connection between bacterial virulence and strain metabolism. However, the present study has several limitations. First, only two strains (one VISA strain XN108 and an MSSA strain Newman) were tested, and more strains are needed to determine the role of WalK(S221P)-MgrA in promoting capsule production in *S. aureus*. Second, WalKR may have various mutations except for WalK(S221P), and other mutations could play similar roles. More mutations like WalK(G223D), WalK(V381I), and WalR(K208R) are needed to further confirm the roles of WalKR in controlling *S. aureus* capsule production. Third, whether blocking WalKR-MgrA affects *S. aureus* virulence is not determined. The in vivo experiments are important for the development of alternative anti-virulence factor therapies, and we will perform these experiments in future.

## Figures and Tables

**Figure 1 microorganisms-13-00502-f001:**
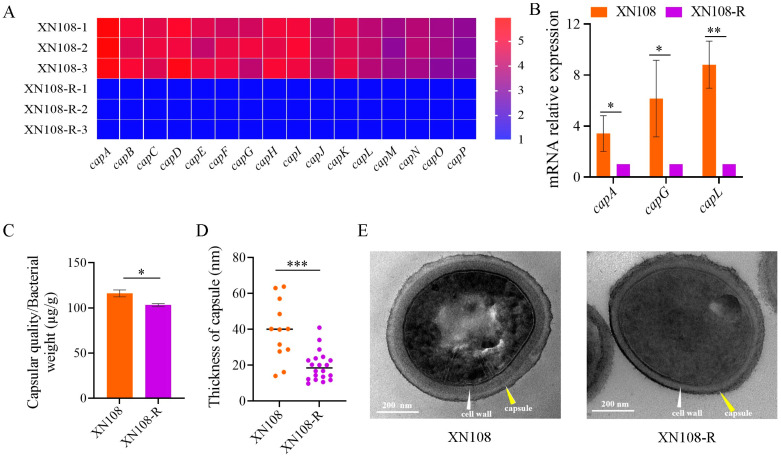
WalK(S221P) mutation promotes the capsule production of *S. aureus*. (**A**) The heat map showing the transcriptomic analysis of 16 *cap* genes in *S. aureus* XN108 versus XN108-R. 1, 2, and 3 represent the three biologically repeated samples. (**B**) RT-qPCR detection of *cap* genes *capA*, *capG*, and *capL* in XN108 and XN108-R. The *gyrB* was used as a reference gene for normalization. The data are expressed as the mean ± standard derivation (SD, *n* = 3). (**C**) Quantitative analysis of capsular polysaccharides with the phenol–sulfuric acid method. The data are expressed as the mean ± SD (*n* = 3). (**D**) Capsule thickness comparison. Data were expressed as mean ± SD (XN108, *n* = 12; XN108-R, *n* = 20). (**E**) Representative TEM images of XN108 and XN108-R. The cell wall and capsule are indicated by white and yellow triangles, respectively. Statistical significance in (**B**–**D**) was calculated by Student’s *t*-test; * represents *p* < 0.05, ** indicates *p* < 0.01, and *** shows *p* < 0.001.

**Figure 2 microorganisms-13-00502-f002:**
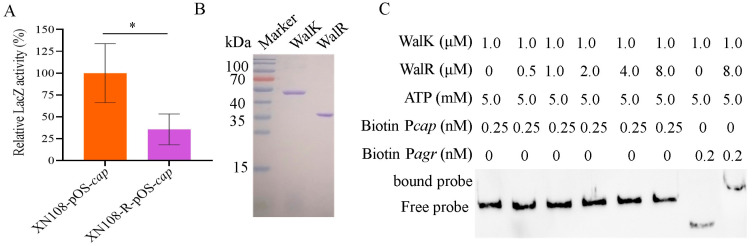
WalK(S221P) mutation indirectly controls *cap* expression. (**A**) β-Galactosidase assay. The pOS-*cap* reporter plasmid was transformed into XN108 and XN108-R, and the LacZ activity was detected. The data are indicated as mean ± SD (*n* = 3). Statistical significance was calculated by Student’s *t*-test, * represents *p* < 0.05. (**B**) SDS-PAGE analysis of the purified WalK and WalR. (**C**) EMSA showing the DNA-binding ability of WalK-activated WalR to the promoter DNA fragment of the *cap* gene (P*cap*). The promoter DNA fragment of *agr* (P*agr*) was used as a positive control.

**Figure 3 microorganisms-13-00502-f003:**
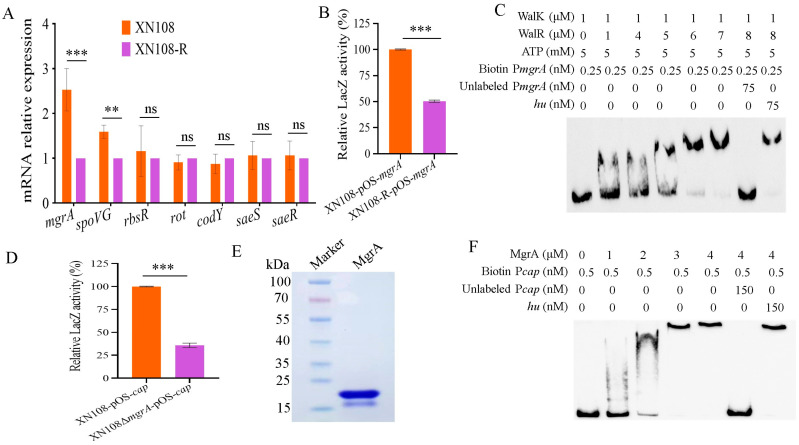
WalK(S221P) mutation increases *cap* expression through MgrA. (**A**) RT-qPCR detection of the regulator genes in XN108 and XN108-R. The *gyrB* gene was used as a reference for normalization. The data are expressed as mean ± SD (*n* = 3). (**B**) β-Galactosidase assay. The pOS-*mgrA* reporter plasmid was transformed into XN108 and XN108-R, respectively, and the LacZ activity was detected. The data are presented as mean ± SD (*n* = 3). (**C**) The EMSA of WalK-activated WalR with the biotin-labelled *mgrA* promoter DNA probe (P*mgrA*). The unlabelled probe was used as the specific competitor, while the unlabelled partial fragment of *hu* ORF region was used as the non-specific competitor. (**D**) β-Galactosidase assay. The pOS-*cap* plasmid was transformed into XN108 and XN108Δ*mgrA*, respectively, and the LacZ activity was detected. Data are presented as mean ± SD (*n* = 3). (**E**) SDS-PAGE analysis of the purified MgrA proteins. (**F**) The EMSA of MgrA with the biotin-labelled *cap* gene promoter DNA probe (P*cap*). Statistical significance in (**A**,**B**,**D**) was calculated by Student’s *t*-test; ns shows no significance, ** indicates *p* < 0.01, and *** represents *p* < 0.001.

**Figure 4 microorganisms-13-00502-f004:**
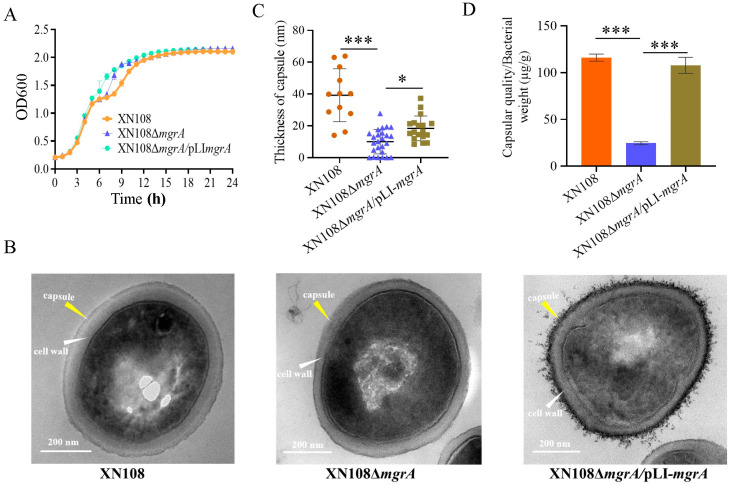
Deletion of *mgrA* reduces capsule production in *S. aureus*. (**A**) Growth curves of XN108 and its derivatives. The data represent the mean ± SD of each time point (*n* = 3). (**B**) Representative TEM images of XN108 and its derivatives. The cell wall and capsule are indicated. (**C**) Capsular thickness of XN108 and its derivatives. (**D**) Capsular polysaccharide quantification with the phenol–sulfuric acid method. Statistical significance was calculated by Student’s *t*-test; * indicates *p* < 0.05 and *** represents *p* < 0.001.

**Figure 5 microorganisms-13-00502-f005:**
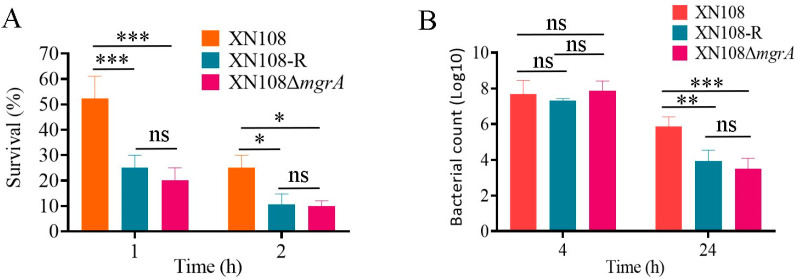
WalK(S221P) mutation and MgrA promotes the intracellular survival of *S. aureus*. (**A**) Whole blood killing. *S. aureus* XN108 and its derivatives were cultured in the BHI broth overnight. On the next day, bacterial cells were collected and co-cultured with mouse whole blood for 1 or 2 h at 37 °C. The survival rate of *S. aureus* cells was calculated after bacterial counting. (**B**) Bacterial count of XN108 and its derivatives 4 and 24 h postphagocytosis by RAW264.7 macrophages. The data represent the mean ± SD (*n* = 3). The statistical significance was calculated by two-way ANOVA. * *p* < 0.05, ** *p* < 0.01, *** *p* < 0.001, and ns indicates no significance.

**Figure 6 microorganisms-13-00502-f006:**
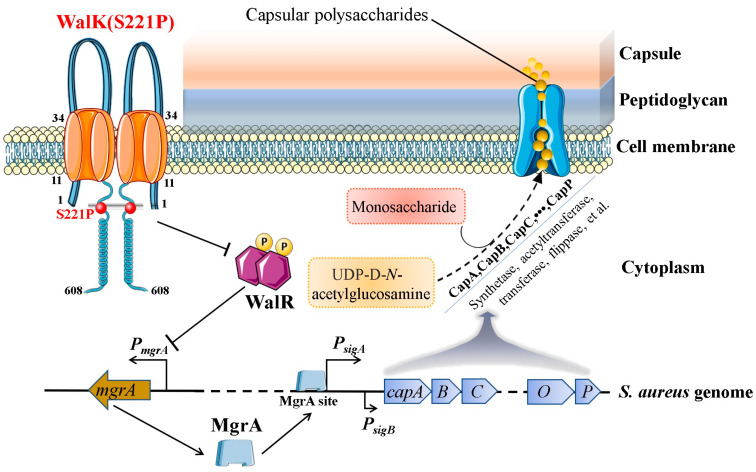
The schematic diagram showing the role of the WalK(S221P) mutation in promoting *S. aureus* capsule production through an MgrA-dependent pathway. The WalK(S221P) mutation in the 3D schematic models was indicated by red balls.

**Table 1 microorganisms-13-00502-t001:** Strains and plasmids used in this study.

Strains/Plasmids	Characteristics	Source/Reference
strains		
*S. aureus* strain		
XN108	A vancomycin-intermediate and methicillin-resistant strain with WalK(S221P) mutation carrying CP8 capsule	[26]
XN108-R	XN108 derivative with WalK(S221P) mutation recovered	[26]
Newman	MSSA strain carrying CP5 without WalK(S221P) mutation	[26]
K-Newman	Newman derivative with WalK(S221P) mutation introduced	[26]
XN108Δ*mgrA*	XN108 derivative with seamless deletion of *mgrA* gene	This work
XN108Δ*mgrA*/pLI-*mgrA*	XN108Δ*mgrA* complemented with the *mgrA* gene	This work
XN108-pOS-*mgrA*	XN108 derivative with plasmid pOS-*mgrA*	This work
XN108-R-pOS-*mgrA*	XN108-R derivative with plasmid pOS-*mgrA*	This work
XN108-pOS-*cap*	XN108 derivative with plasmid pOS-*cap*	This work
XN108Δ*mgrA*-pOS-*cap*	XN108Δ*mgrA* derivative with plasmid pOS-*cap*	This work
*E. coli* strain		
DH5α	Cloning host for maintaining recombinant plasmids	Biomed
BL21(DE3)	Expression host for exogenous protein production	Biomed
plasmids		
pBT2	Shuttle vector, temperature sensitive, Amp^R^, and Cm^R^	[31]
pLI50	Expression vector, Amp^R^, and Cm^R^	[31]
pOS1	Reporter vector with lacZ coding sequence for the β-galactosidase assay, Cm^R^	[31]
pET28a	Expression vector, Kan^R^	[31]

Cm^R^, chloramphenicol resistant; Kan^R^, kanamycin resistant; Amp^R^, ampicillin resistant.

## Data Availability

The RNA-seq data in this study are deposited in the GEO DataSets under the ID code of GSE127706.

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
