# Peer review of "WalK(S221P) Mutation Promotes the Production of Staphylococcus aureus Capsules Through an MgrA-Dependent Pathway"

_microorganisms, 2025, doi:10.3390/microorganisms13030502_

Round 1

Reviewer 1 Report

Comments and Suggestions for Authors

The authors present a detailed study about the role of of WalK(S221P) mutation in promoting capsule production of S. aureus through an MgrA-dependent pathway. The manuscript has scientific soundness and will be of interest to those involved in S. aureus research field. The work was performed applying state of the art methodologies to support each conclusion. Just some suggestions before its acceptance for publication.

1.- The description of several sections in methodology needs to be improved including more information and not just making the reference to a previous work.

2.- An integrative scheme about the role of WalK(S221P) mutation in promoting capsule production including MgrA would be useful.

3.- In the same context, a 3D model generation of the structure of the mutant protein will be interesting to compare with the wild type to try to gain information to explain the data obtained.

Author Response

Point-by-point response to the respected Reviewer: 1

We appreciate all constructive and professional comments & suggestions given by the Reviewer’s. We have studied the comments carefully and made a revision to address all your questions within the manuscript with yellow-highlighted red texts. Here, we would like to responded your questions point-by-point as following:

Comments:

The authors present a detailed study about the role of WalK(S221P) mutation in promoting capsule production of S. aureus through an MgrA-dependent pathway. The manuscript has scientific soundness and will be of interest to those involved in S. aureus research field. The work was performed applying state of the art methodologies to support each conclusion. Just some suggestions before its acceptance for publication.

Response: Thank you so much for the professional comments. These comments are vital and helpful to improve the quality of our manuscript. We analyzed your comments critically and revised the manuscript as possible as we can.

1.- The description of several sections in methodology needs to be improved including more information and not just making the reference to a previous work.

Response: Thank you so much for the reasonable comments. To improve the methodology, we added detailed information for the Methods in our revised manuscript with yellow-highlighted red texts, including Bacterial Strains (lines 91–93,103), RNA-seq (lines 109–116), RT-qPCR (lines 119–121), and TEM Observation (lines 154–163).

2.- An integrative scheme about the role of WalK(S221P) mutation in promoting capsule production including MgrA would be useful.

Response: It is a good idea. We included the model diagram of Figure 6 in the conclusion section of our revised manuscript.

3.- In the same context, a 3D model generation of the structure of the mutant protein will be interesting to compare with the wild type to try to gain information to explain the data obtained.

Response: Thank you for your professional suggestion. A 3D model diagram was generated and included in Figure 6 of our revised manuscript.

On behalf of all co-authors, I thank you in advance for your time, effort, and consideration.

Yours sincerely,

Xiancai Rao, Ph.D,

Department of Microbiology,

Army Medical University (Third Military Medical University)

Gaotanyan St., Shapingba District, Chongqing, 400038, P. R. China

Tel: 86-23-68771350

E-mail: xcrao@tmmu.edu.cn, or raoxiancai@126.com

Reviewer 2 Report

Comments and Suggestions for Authors

In the present study, the authors aimed to demonstrate that the production of Staphylococcus aureus capsules is induced by WalK(S221P) mutation through an MgrA-dependent pathway. Explaining bacterial virulence and the factors involved in it is essential due to the progressive increase of S. aureus resistance to various antibiotics; further antibacterial drugs could target these mechanisms.

The following comments are available below:

  1. Abstract.

In the present version is blurred. Please organize it as follows: Background/Objective, Materials and Methods, Results, and Conclusions.

Please revise the keywords and include "virulence" after S. aureus—an essential term to highlight the study's importance. "Capsular polysaccharides" is the third important keyword, followed by the others.

2. Introduction

Please focus on the role of capsular polysaccharides' expression in S. aureus virulence and show the factors that can influence it in both directions (increase/damage).

Please describe the significant role of the 2 players (WalK(S221P) mutation and MgrA) in S. aureus capsular polysaccharides sequence regulation. The authors should include the phrases from lines 237-238, 263-265, 282-284, 326-329 and 332 here. 

Please show the aim of the present study and the hypothesis more precisely. 

3. Materials and Methods:

Generally, all are described enough, with suitable references.

Line 83: Table S1 is relevant. Please put it in the MS text.

The reviewer understands that the present study is available for VISA S. aureus strains XN108. If this observation is correct, correct, please make it more explicit and mention it in the title.

4. Results:

The Figures are of high quality.

Please put each figure after the first mention in the MS text section and extensively present the results using the values of the obtained variable parameters.

5. Discussions

Please show the reason for such a study design.

When discussing each study step, please show the importance of each experiment and compare the original data with those from the scientific literature.

Please discuss the limitations of the present study.

Finally, please show the potential applications of the present research in medical practice. 

6. Conclusions:

Please show further research directions.

Author Response

Point-by-point response to the respected Referee: 2

We appreciate all constructive and professional comments & suggestions given by the Reviewer’s. We have studied the comments carefully and made a revision to address all questions within the manuscript with yellow-highlighted red texts. Here, we would like to responded your questions point-by-point as following:

Comments:

In the present study, the authors aimed to demonstrate that the production of Staphylococcus aureus capsules is induced by WalK(S221P) mutation through an MgrA-dependent pathway. Explaining bacterial virulence and the factors involved in it is essential due to the progressive increase of S. aureus resistance to various antibiotics; further antibacterial drugs could target these mechanisms.

The following comments are available below:

  1. In the present version is blurred. Please organize it as follows: Background/Objective, Materials and Methods, Results, and Conclusions.

Please revise the keywords and include "virulence" after S. aureus—an essential term to highlight the study's importance. "Capsular polysaccharides" is the third important keyword, followed by the others.

Response: Thank you so much for your professional comments and suggestions. According to the Instructions for Authors of the target Journal, the abstract should be a total of about 200 words maximum, and the abstract should be a single paragraph and should follow the style of structured abstracts, but without headings. To address your question, we revised the abstract and try our best to follow the style of structured abstracts.

   The keywords were also revised according to your helpful suggestion.

  1. Introduction

Please focus on the role of capsular polysaccharides' expression in S. aureus virulence and show the factors that can influence it in both directions (increase/damage).

Please describe the significant role of the 2 players (WalK(S221P) mutation and MgrA) in S. aureus capsular polysaccharides sequence regulation. The authors should include the phrases from lines 237-238, 263-265, 282-284, 326-329 and 332 here. 

Please show the aim of the present study and the hypothesis more precisely. 

Response: Thank you for the valuable comments and suggestions. The Introduction section was rewritten to focus on the role of CP production in S. aureus virulence. The factors that can influence CP biosynthesis, including the positive regulators such as MgrA, SpoVG, and RbsR and negative modulators like Rot, CodY, and SaeSR that control the expression of cap gene cluster were provided. The roles of WalK(S221P) mutation and MgrA in CP regulation in S. aureus were also added.

The phrases described in lines 237–238, 263–265, and 282–284 of the previous version of manuscript were moved to the Introduction section according to your valuable suggestions. The aim of the present study was also supplemented in the revised version of manuscript.

  1. Materials and Methods:

Generally, all are described enough, with suitable references.

Line 83: Table S1 is relevant. Please put it in the MS text.

The reviewer understands that the present study is available for VISA S. aureus strains XN108. If this observation is correct, correct, please make it more explicit and mention it in the title.

Response: Thank you for the important comments and suggestions. Table S1 was put in the MS text as Table 1 in the revised manuscript. Yes, the present study is firstly started fromVISA S. aureus strain XN108, however, we tested the function of WalK(S221P) mutation in MSSA strain Newman. Therefore, only mention of VISA XN108 in the title may not be suitable, and we would like to keep the original title.

  1. Results:

The Figures are of high quality.

Please put each figure after the first mention in the MS text section and extensively present the results using the values of the obtained variable parameters.

Response: Thank you for the valuable comments and suggestions. All figures were put after the first mention in the MS text, and the results were presented with the obtained values as possible as we can.

  1. Discussions

Please show the reason for such a study design.

When discussing each study step, please show the importance of each experiment and compare the original data with those from the scientific literature.

Please discuss the limitations of the present study.

Finally, please show the potential applications of the present research in medical practice. 

Response: These are very nice comments and suggestions. The study design was provided. The importance of certain experiments and result comparison were added as possible as we can. The potential applications of the present research in medical practice were also provided.

  We discussed the limitations of the present study to address your concern. However, we put them in the Conclusion section, where they could be combined with further research directions.

  1. Conclusions:

Please show further research directions.

Response: Thank you so much for the professional suggestions. The further research directions were provided in the revised manuscript as possible as we can.

On behalf of all co-authors, I thank you in advance for your time, effort, and consideration.

Yours sincerely,

Xiancai Rao, Ph.D,

Department of Microbiology,

Army Medical University (Third Military Medical University)

Gaotanyan St., Shapingba District, Chongqing, 400038, P. R. China

Tel: 86-23-68771350

E-mail: xcrao@tmmu.edu.cn, or raoxiancai@126.com